# Rutin Protects Fibroblasts from UVA Radiation through Stimulation of Nrf2 Pathway

**DOI:** 10.3390/antiox12040820

**Published:** 2023-03-28

**Authors:** Elisabetta Tabolacci, Giuseppe Tringali, Veronica Nobile, Sara Duca, Michela Pizzoferrato, Patrizia Bottoni, Clementi Maria Elisabetta

**Affiliations:** 1Dipartimento di Sanità Pubblica e Scienze della Vita, Sezione di Medicina Genomica, Università Cattolica del Sacro Cuore, Largo F. Vito 1, 00168 Rome, Italy; elisabetta.tabolacci@unicatt.it (E.T.);; 2Fondazione Policlinico Universitario A. Gemelli IRCSS, 00168 Rome, Italy; 3Pharmacology Section, Department of Health Care Surveillance and Bioethics, Università Cattolica del Sacro Cuore, Largo F. Vito 1, 00168 Rome, Italy; 4UOC Genetica Medica, Fondazione Policlinico Universitario A. Gemelli IRCCS, 00168 Rome, Italy; 5Dipartimento di Scienze Biotecnologiche di Base, Cliniche Intensivologiche e Perioperatorie, Università Cattolica del Sacro Cuore, Largo F. Vito 1, 00168 Rome, Italy; 6Istituto di Scienze e Tecnologie Chimiche “Giulio Natta” SCITEC-CNR, Largo Francesco Vito 1, 00168 Rome, Italy

**Keywords:** rutin, UVA-induced damage, Nrf2 pathway, oxidative stress, mitochondrial function

## Abstract

This study explores the photoprotective effects of rutin, a bioflavonoid found in some vegetables and fruits, against UVA-induced damage in human skin fibroblasts. Our results show that rutin increases cell viability and reduces the high levels of ROS generated by photo-oxidative stress (1 and 2 h of UVA exposure). These effects are related to rutin’s ability to modulate the Nrf2 transcriptional pathway. Interestingly, activation of the Nrf2 signaling pathway results in an increase in reduced glutathione and Bcl2/Bax ratio, and the subsequent protection of mitochondrial respiratory capacity. These results demonstrate how rutin may play a potentially cytoprotective role against UVA-induced skin damage through a purely antiapoptotic mechanism.

## 1. Introduction

The risks and damages induced by excessive exposure to UV radiation, especially the solar one, have been the object of careful evaluations and studies, especially in recent decades. Prolonged exposure to solar UV radiation can cause acute and chronic harmful effects to the skin, eyes, connective tissue, blood vessels, and, in the most serious cases, skin tumors. In this regard, among the chronic effects of major health relevance, there are some neoplastic forms such as melanoma, basal cell carcinoma (BCC), and squamous cell carcinoma (SCC). These types of cancers often appear on areas of the skin that are not adequately protected from solar radiation [1,2,3,4].

There are different types of ultraviolet radiation, which have different effects depending on the wavelength: the longer the wavelength is, the greater is the ability of the rays to penetrate the atmosphere and reach the deep layers of the skin tissue [5]. In this context, UVA rays (315–400 nm), characterized by a constant intensity throughout the year, represent around 95% of the UV radiation capable of reaching dermal fibroblasts, also causing long-term damage [6,7]. Exposure to UVA rays leads to (i) an increased oxidative stress in fibroblasts, (ii) a decreased secretion and synthesis of matrix proteins, (iii) cellular dysfunction, and, finally, (iv) apoptosis: these processes first promote skin aging and later promote the predisposition to skin diseases [7,8,9,10,11]. The reactive oxygen species (ROS) produced (after UVA exposure) interact with lipid-rich membranes, enzymes, and nuclear DNA by changing their structures and interfering with their functions. In addition, the accumulation of ROS in cells leads to the initiation of cellular reactions with dysfunctions of mitochondria activities. Therefore, ROS accumulation in skin cells represents the main cause of photodamage, photoaging, and photocarcinogenesis [12,13,14,15]. At the molecular level, it is also known that UVA-generated ROS lead to a decreased expression of erythroid nuclear factor 2-like 2 (Nrf2), a marker of oxidative stress [16,17,18]. Nrf2 is a key regulator in the protection of skin cells from oxidative stress and UVA radiation [19]. In fact, increasing intracellular Nrf2 leads to the maintenance of the cell’s reducing potential, the decrease in apoptotic genes, and the maintenance of mitochondrial integrity resulting in improved cell viability [20,21].

The use of natural substances that can protect cells from oxidative stress has now become the subject of numerous studies, either in the form of nutraceuticals or used topically as protectants. In our study, we planned to analyze the effect of rutin, a flavonoid naturally occurring in many foods, especially buckwheat, apricots, cherries, grapefruit, plums, and oranges, and it was found in high amounts in the waste from wine production from grapes. It is often used in patients with capillary fragility, varicose veins, and hematomas, but recent papers have shown an important antioxidant effect [22,23,24]. Therefore, our experimental design aimed at studying the effect of rutin on fibroblast cultures exposed to UVA radiation, focusing our preliminary/exploratory study on oxidative stress and its mechanisms.

## 2. Materials and Methods

### 2.1. Cells and Treatments

Fibroblasts derived from a healthy control male (CTRL, coded CTRL1) of 31 years old were employed. A fibroblast cell culture was established after around 1 week from the date of biopsy. Skin biopsy was performed at the left leg (around 10 cm from the ankle) and was obtained after a signed Informed Consent (prot. N. 9917/15 and prot.cm 10/15 of Ethics Committee at the Catholic University of Rome). The cell culture was grown in DMEM medium (Sigma Aldrich, St. Louis, MO, USA), supplemented with 10% fetal bovine serum (FBS), 1% penicillin/streptomycin, and 2.5% L-glutamine at 37 °C with 5% CO_2_. For the subsequent experiments, cells were seeded with 80% of confluency. UVA exposure was produced using a lamp (Vilber Lourmat VL-62C Power 6W; Vilber Lourmat Deutschland GmbH, Eberhardzell, Germany) at 365 nm, and was placed 10 cm from the source for 1, 2, 3, 4, and 5 h at an intensity of ~0.06 J/cm^2^/s. To minimize radiation uptake by the medium, the cells were kept in PBS during exposure, and immediately after exposure, the culture medium was replaced and the cells were put in an incubator for 24 h before proceeding to the different assays. Control cells (unirradiated) were maintained for the same period under the same experimental condition.

Considering the results obtained from the cell viability curve, we chose to continue the subsequent experiments by exposing the cells to UVA radiation for 1 and 2 h.

Rutin (purchased from Sigma) was dissolved in DMSO to a final concentration of 10 mM and then, before use, the solution was diluted in PBS to the desired concentrations. Rutin was added to the cells 24 h before UVA exposure. To determine a dose–response curve, we evaluated the cell viability after UVA exposure for 1 and 2 h, using rutin at 5, 10, 20, and 25 µM. Based on the data obtained, we chose 10 µM as the optimal concentration.

All experiments were also conducted in the presence of DMSO alone at the highest concentration used to dissolve 25 μM rutin. No difference was found compared with control cells.

### 2.2. Cell Viability

Cell viability was assessed with the MTS assay (Promega srl, Padua, Italy), according to the manufacturer’s instructions. Briefly, after UVA exposure and various experimental treatments, MTS reagent was added to cells and plated in 96-well plates at a cell density of 2 × 10^4^ cells/well, reaching around 80% of confluency. The assay provides a sensitive measurement of the normal metabolic state of cells, reflecting early changes in cellular redox homeostasis. Intracellular soluble formazan produced by the cell reduction of MTS is proportional to the number of live cells, and it was measured by recording the absorbance of each well of the plate with a plate reader at 490 nm. The cellular viability was expressed as % compared with control cells.

### 2.3. ROS Measurement

An intracellular ROS detection kit containing 2′,7′-dichlorofluorescein diacetate (DCF-DA) was used to measure the formation of intracellular ROS (Abcam, Cambridge, UK). Briefly, DCF-DA was applied in accordance with the manufacturer’s instructions to fibroblasts grown in 96-well microplates (20,000 cells per well) and subjected to various experimental settings. When ROS are present, DCF-DA, an originally non-fluorescent molecule, is oxidized to DCF, a highly fluorescent chemical. A CytoFluor multiwell plate reader (Victor3-Wallac-1420; PerkinElmer, Waltham, MA, USA) was used to measure the fluorescence at 485/538 nm of excitation/emission. The amount of ROS produced was proportional to the emitted fluorescence and measured by fluorescence intensity.

### 2.4. Nrf2 Detection Assay

A cell-based colorimetric ELISA kit was used to measure the intracellular levels of Nrf2 (LSBio, LifeSpan Biosciences; Seattle, WA, USA). In a 96-well plate, fibroblasts were seeded at a density of 20,000 cells per well. Following the UVA exposure and treatment, cells were fixed with 4% formaldehyde. Finally, the plate was incubated at 4 °C overnight with the addition of quenching solution, blocking buffer, and the primary antibodies (against-Nrf2—a rabbit polyclonal antibody; against-GAPDH—a mouse monoclonal antibody, employed as an internal positive control and used for normalization). The samples were examined with a microplate reader at a wavelength of 450 nm after the addition of the peroxidase-conjugated secondary antibody. The colorimetric results were provided as a percentage compared with the untreated control and normalized as the OD450 of the Nrf2/GAPDH ratio.

### 2.5. Total and Reduced Glutathione Assay

Total (GSH + GSSG) and reduced glutathione (GSH) levels were assessed using a colorimetric Assay Kit (ab239709, Abcam, Cambridge, UK). Briefly, fibroblasts seeded in Petri dishes at 80% confluence were pre-treated with 10 µM rutin for 24 h and then exposed to UVA irradiation for 1 and 2 h. After irradiation, cells were lysed by adding the lysis buffer provided in the kit and centrifuged at 14,000× *g* for 10 min. Total glutathione and reduced fraction were measured in the supernatants after normalization for mg/protein (see below). The glutathione assay is based on the reaction between DTNB (glutathione substrate) and glutathione that generates 2-nitro-5-thiobenzoic acid, which has a yellow color, and therefore, the concentration of GSH is determined by measuring the absorbance at 412 nm. To detect only the reduced form of glutathione, glutathione reductase is omitted from the assay. Standard curves (substrates provided by the kit) were used to calculate the absolute value of total glutathione (ng/mL) and glutathione in reduced form (µg/mL).

Protein concentration was determined with a Protein Assay (Bio-Rad, Hercules, CA, USA) in 96-well microplates using a calibration curve for BSA.

### 2.6. Measurement of Iron Level

Total iron levels in the cytoplasm of fibroblasts treated according to the experimental design previously described were assessed with the Iron Assay kit (Abcam, ab83366), according to the manufacturer’s instructions. Briefly, 2 × 10^6^ cells were lysed in 250 μL of assay buffer provided by the kit, and 50 μL of lysates, at the same protein concentration, was placed in 96 wells plate. Calibration standard curves were placed in the same plate. For the determination of total iron (II + III), an iron reducer was added to the wells so that all ferric iron was reduced to ferrous. The plate was incubated at 37 °C for 30 min, then the Iron Probe was added and incubated for an additional 60 min. Free iron (II) reacts with the Iron Probe to produce a stable-colored complex, and it was evaluated immediately on a colorimetric microplate reader (OD = 593 nm). The iron(II) and total iron content of the test samples can be directly determined from the standard curve (standard provided by the kit). Iron(III) was calculated as total iron-iron(II). Values were presented as nmoles present in the 50 μL of the cellular lysates.

### 2.7. Bax and Bcl2 Detection Assay

Bax and Bcl2 proteins were used to assess the degree of apoptosis of fibroblasts exposed to UVA radiation for 1 and 2 h with and without pretreatment with 10 µM rutin. Intracellular concentrations of Bax and Bcl2 proteins were determined using a colorimetric cell ELISA kit from Assay Biotechnology (Sunnyvale, CA, USA). Cells were seeded in a 96-well plate at a density of 20,000 cells/well and treated as previously reported. Twenty-four hours after treatment, primary antibodies (rabbit polyclonal against Bax, rabbit polyclonal against Bcl2, and mouse monoclonal against GADPH, the latter used as housekeeping) were added to the cells, followed by peroxidase-coupled secondary antibodies.

A microplate reader was used to measure the samples at 450 nm, and the ODs of Bax and Bcl2 were normalized for housekeeping. To establish an arbitrary anti-apoptotic index, we expressed the ratio of OD of Bcl2 (anti-apoptotic protein) vs. OD of Bax (apoptotic protein).

### 2.8. High-Resolution Respirometry (HRR)

Respiration in intact fibroblasts was monitored with high-resolution respirometry (Oroboros Oxygraph-2k, Innsbruck, Austria) operating at 37 °C with a 2 mL chamber volume [25]. Cellular respiration experiments were carried out in two O2k chambers operated in parallel after calibration of the oxygen sensors at air saturation and an instrumental background correction. Calibration with air-saturated medium was performed immediately before the oxygen flux measurement was taken. The data acquisition and analysis were carried out using DatLab software (Oroboros Instruments). Fibroblasts were seeded in a Petri dish with 80% of confluency. After seeding, cells were pre-treated with 10 µM of rutin for 24 h and then were exposed to UVA for 1 and 2 h. After irradiation, fibroblasts were trypsinized, counted, resuspended in supplemented DMEM medium to a final concentration of 1 × 10^6^ cells/mL, added to each Oxygraph Chamber (Chamber A and Chamber B), and thereafter investigated using a phosphorylation control protocol [26]. The experiments began with the measurement of the basal oxygen consumption rate (OCR), followed for about 10 min, until a steady state level was obtained (Basal OCR). ATP synthase was inhibited by the addition of oligomycin (2 µg/mL) at each chamber to detect the OCR from Proton Leak. The maximal respiration capacity (Maximal OCR) was obtained by the addition of small volumes of the uncoupler carbonyl cyanide-4-(trifluoromethoxy) phenylhydrazone (FCCP, 0.25 μM FCCP/step) and by the instantaneous observation of its effect on cellular respiration in the uncoupled state. Cell respiration was then measured in the presence of 0.5 μM rotenone, which selectively inhibits Complex I, and then in the presence of 2.5 μM antimycin A, which inhibits Complex III, to estimate Residual OCR. In addition to instrumental background, the mitochondrial respiration was corrected for the oxygen flux due to residual OCR.

### 2.9. Caspase-3 Activity

The supernatants obtained, as reported for the GSH assay, were also used to evaluate the caspase-3 activity, quantified using a specific colorimetric kit (Caspase-3 Assay Kit Abcam ab39401). Briefly, cell lysates, at the same protein concentration, were mixed with an equal volume of reaction buffer containing 10 mM DTT and 200 µM DEVD-p-NA substrate. The mixture was incubated at 37 °C for 2 h. Optical density was measured at 400 nm with a microplate reader. The number-fold increase in caspase-3 activity was determined by comparing the results of the samples (irradiated and/or treated cells) with the level of the untreated and unirradiated control. Data were expressed as the percentage of activity compared with the control cells.

### 2.10. Statistical Analysis

Each experiment was replicated at least three times with up to eight replicates per group. Results were displayed as means ± SEM. Data were analyzed by one-way ANOVA with the Newman–Keuls post hoc test by using PrismTM software (GraphPad, San Diego, CA, USA). Statistical analysis for HHR was performed using the Newman–Keuls Multiple Comparison Test. The level of significance was set at *p* ≤ 0.05.

## 3. Results

We first tested the response of fibroblasts exposed to different times of UVA irradiation by evaluating their viability. After a time titration curve (1, 2, 3, 4, and 5 h) of UVA exposure, fibroblasts showed a significant loss of viability of 35 and 50% after 1 and 2 h, respectively (Figure 1), compared with control cells (blue line), which were not treated with UVA, but kept under a hood outside the incubator. From the 3rd hour onward, cell death was also observed in control cells and therefore not attributable exclusively to the action of UVA. In light of these results, we chose 1 and 2 h stay under the lamp as the optimal time of UVA exposure.

After evaluating the UVA exposure conditions, we sought to identify the optimal concentration of rutin, the molecule we chose as a potential protective agent. We then pretreated the fibroblasts with different concentrations of rutin (5, 10, 15, and 25 µM) and, after 24 h, the cells were subjected to UVA irradiation for 1 and 2 h. As shown in Figure 2, which reports cell viability, pretreatment with rutin had no effect on cells not exposed to UVA (blue bars), while it protected cells from UVA-dependent cytotoxicity. This result was evident even at low concentrations (5 µM): after exposure, there were 15% and 32% increases in viability at 1 h and 2 h, respectively, compared with the control (UVA-exposed cells without rutin). The result was even more pronounced at 10 µM, when the increase in cell viability after 1 h of exposure was 20% and 45% compared with controls. These data demonstrated the protective effect of rutin on UVA-exposed cells, and in order to identify the molecular mechanisms underlying this phenomenon, we chose a concentration of 10 µM for subsequent experiments.

To investigate the molecular mechanisms underlying UVA damage and the protective effect of rutin, we measured the presence of ROS after administration of 10 µM rutin and the subsequent exposure to UVA radiation for 1 and 2 h, respectively (Figure 3). ROS were measured as fluorescence intensity and expressed as a percentage compared with unirradiated, non-rutin-treated cells (arbitrarily set as 100%). As can be seen, in cells without rutin, the increase in ROS was 20% after 1 h of exposure and tripled after 2 h. In contrast, in the rutin-pretreated cells, the oxidation levels were similar to those of the unexposed cells, showing a significant reduction in ROS compared with the untreated cells.

As it is known that increased ROS within cells lead to an immediate increase in Nrf2, which seeks to counteract cellular oxidation, we measured this protein in fibroblasts treated according to the previous experimental protocol. In Figure 4, we report Nrf2 values in cells unexposed and in cells exposed for 1 and 2 h to UVA irradiation, with and without pre-treatment with 10 µM of rutin. Even in cells not exposed to radiation, rutin evidently resulted in increased Nrf2 levels (reaching threefold values compared with control) and, after UVA exposure, a 3.5- and 4-fold increase at 1 and 2 h of radiation exposure, respectively. Note that in fibroblasts not pre-treated with rutin, the Nrf2 level did not significantly change after 1 h of exposure, whereas a slight decrease occurred after 2 h of UVA irradiation without reaching statistical significance.

As Nrf2 is known to induce an increase in the antioxidant defenses of cells through several mechanisms among which the increase in the antioxidant potential of cells emerges, the next step was to assess the glutathione in the cytosol of treated fibroblasts according to our experimental design. Table 1 shows the values of total glutathione expressed in ng/μL and reduced glutathione in μg/μL. As can be seen, there were no significant changes in total glutathione in the various samples, demonstrating that both UVA and rutin treatment did not change glutathione expression. In contrast, the reduced form of glutathione (GSH) was significantly decreased following UVA treatment and significantly increased in cells pre-treated with rutin. These data indicate that fibroblasts, in order to counteract the UVA-induced increase in ROS, consumed the cellular antioxidant stores resulting in GSH depletion, which decreased from 2.30 (μg/μL) to 1.36 and 1.07 after 1 and 2 h of irradiation, respectively. In contrast, pre-treatment with rutin doubled the amount of GSH before radiation exposure (which increased from 2.20 μg/μL to 5.20 µg/µL) and, following UVA rays, kept the antioxidant potential of the cells high by counteracting ROS formation, as seen.

The increase in Nrf2 and GSH is thus a cellular defense mechanism that rutin induces to counteract UVA damage. As recent work has shown that rutin acts as a cytoprotective agent through the induction of Nrf2 by both preventing ferroptosis and apoptosis, we further investigated the molecular basis of the phenomenon observed in our experimental system. First, we assessed intracellular levels of total iron (as a marker of ferroptosis) and of the two forms, oxidized Fe(II) and reduced Fe(III), in fibroblasts irradiated with UVA for 1 and 2 h in the presence and absence of 10 µM of rutin. As can be seen from Figure 5, no significant changes in total iron and ferrous and ferric iron were observed in fibroblasts exposed to UVA or treated with rutin: this excludes that the cytotoxicity shown by UVA was induced by ferroptosis.

Next, to assess the presence of apoptotic mechanisms, we measured the amount of Bcl2 and Bax in the cells and reproduced the value of the “antiapoptotic index” by expressing the ratio of Bcl2 (antiapoptotic) to Bax (pro-apoptotic) in fibroblasts exposed to UVA radiation (for 1 and 2 h) with and without pre-treatment with rutin (Figure 6). As evident in the radiation-exposed cells, Bcl2/Bax decreased by 20% (UVA 1 h) and 25% (UVA 2 h), highlighting a prevalence of pro-apoptotic mechanisms, whereas the presence of rutin practically maintained the ratio as in cells not exposed to UVA.

To assess the protective effects of rutin on the respiratory capacity of human dermal fibroblasts, HRR measurements were performed on cells exposed to UVA radiation (2 h) with and without pre-treatment with 10 µM of rutin. HRR measures, displayed as oxygen flux per cell number, revealed that rutin induced a statistically significant increase in basal respiration of treated fibroblasts compared to control cells. Similarly, a significant growth in basal OCR was measured in fibroblasts exposed to UVA and pretreated with rutin vs. UVA exposed (Figure 7A). Oligomycin-sensitive respiration (Proton Leak), which is induced by inhibiting ATP synthase with oligomycin and corresponds to resting, non-phosphorylating electron transfer, showed a slight trend to increase following rutin treatment, without reaching statistical significance (Figure 7B). Similar to basal OCR, the level of maximal uncoupled respiratory activity (Maximal OCR), recorded in the presence of optimal uncoupler (FCCP) concentrations, was positively influenced by rutin. Notably, maximal OCR, which is a measure of functionality of the mitochondrial respiratory system independently of the cellular energy demand, was significantly increased by about 40% in UVA exposed and rutin-pre-treated cells compared to UVA-exposed cells (Figure 7C). Lastly, the residual OCR, measured upon addition of rotenone and antimycin A, revealed among samples several significant differences in non-mitochondrial oxygen-consuming processes as shown in Figure 7D.

Finally, because all data converged to indicate an inhibition of apoptotic mechanisms by rutin on UVA-exposed cells, caspase-3 activity was measured. It was assumed [27] that caspases regulate the final stages of apoptosis and, in particular, caspase-3 is an “executor” of cell death. Figure 8 shows the values of caspase-3 activity in fibroblasts exposed to UVA radiation with and without rutin pretreatment. As can be seen, exposure to 1 and 2 h of radiation caused an increase by 50 and 100%, respectively, of caspase activity in untreated cells. Pretreatment with rutin, consistent with previous data, did not change caspase activity compared with unirradiated cells.

In conclusion, our results show that fibroblasts irradiated with UVA for 1 and 2 h reach cell death by triggering apoptotic-type mechanisms, and pretreatment with rutin (24 h before exposure) preserves cells from this cell death phenomenon.

## 4. Discussion

Rutin, a naturally occurring flavonoid glycoside in many plant species and/or in the waste of their production cycle (e.g., from the wine production chain), has been shown to exhibit several biological activities, such as antimicrobial, anticarcinogenic, antithrombotic, cardioprotective, and neuroprotective [23,28,29]. Biological actions would seem to be due to its antioxidant, anti-inflammatory, and antiapoptotic properties, capable of protecting the cells from the harmful effects, especially free radicals [30]. For this reason, rutin effects have also been studied on several neurodegenerative diseases, including Alzheimer’s disease (AD), Parkinson’s disease (PD), and Huntington’s disease (HD) [28,30,31]. Furthermore, the rutin in the phospholipid complex is better soluble and permeable than the free rutin, thus allowing its delivery via the skin for the treatment of acute and chronic inflammatory diseases in vivo [32].

In this exploratory work, we exposed to UVA radiation a human fibroblast cell line to obtain an experimental model of photo-induced damage. Ultraviolet radiation triggers a series of chain reactions in exposed cells, that initially determine an increase in intracellular ROS levels, which favor apoptotic processes and consequently cell death. Our previous works, conducted on different cellular models, showed that the increase in ROS levels, consequent to photo-oxidative damage, are linked to the activation of the signaling pathways of the transcription factor Nrf2 [33,34,35]. The results exposed here follow our data previously obtained. In fact, fibroblasts exposed to UVA rays, especially after two hours, show an increase of intracellular ROS levels and of the associated apoptotic molecular events (increase in the Bcl2/Bax ratio), involving the Nrf2 signal transduction pathways [36,37,38,39,40,41,42].

Interestingly, the increase in ROS observed during the two-hour UVA exposure is not associated with an increase in Nrf-2 nuclear levels. Usually, the increase in ROS levels, induced by pro-oxidative stimuli, causes a ready activation of the Nrf-2 pathway, but in the case of UVA rays, the increase in ROS levels is not associated with an immediate increase in Nrf2 nuclear levels. In fact, in the case of photo-oxidative stimuli, intranuclear levels of Nrf2 remain unvaried in the first hours and increase after 3 h of exposure, as already observed in our work [33,43,44] and reported in the literature [45]. The cellular response and nuclear translocation of Nrf2 probably follow different times and modes depending on the chemical and physical properties of the noxious stimuli. Based on the results obtained in our study, it can be hypothesized that GSH, the most important antioxidant in cells, directly interacts with ROS to form oxidized glutathione (GSSG) and that regeneration of oxidized GSH (GSSG) is mediated by an enzyme, glutathione reductase (GSR), which is stimulated by Nrf2. In our experimental system, we see that reduced GSH decreases significantly upon exposure to UVA, a sign of an immediate attempt by the cells to defend themselves against oxidative stress. Subsequently, the decrease in GSH concentration due to UVA-stimulated ROS production causes an alteration of redox homeostasis in cells that can no longer defend themselves against reactive oxygen species.

The mechanism of the short-term activation of the Nrf2 signaling pathway is an important pathway for protection from oxidative damage in skin cells, but in our experimental system, it fails to intervene.

Pretreatment with rutin therefore seems to predispose cells to be more effectively equipped to deal with the damages of photo-exposure and the consequent increase in ROS, given that Nrf2 already increases after 24 h in unexposed cells. This increase also leads to a greater capacity of cells to be reactive, in the short term, against oxidative stress, as demonstrated by the values of reduced GSH. The increase in Nrf2 therefore provides cells with antioxidant potential that facilitates the disposal of reactive oxygen species after an acute insult such as UVA exposure.

Within this framework, it is interesting to underline that rutin can protect fibroblasts from UVA damage both directly as an ROS scavenger and indirectly in the form of rutin “quinone”, stimulating endogenous antioxidant systems mediated by Nrf2 [46]. In fact, rutin is an antioxidant flavonoid, which is quickly oxidized and converted to quinone during its action as an ROS scavenger. This is possible because the chemical structure of rutin presents some isomers Orto and Para-diphenols, which, in conditions of oxidative stress, are converted into the correspondent quinones. The latter easily bind to the thiol-groups present on the Keap-1 protein, facilitating the nuclear translocation of Nrf2 [47,48].

However, whatever the mechanism, activation of the Keap1/Nrf2/Are signaling pathway clearly plays a central role in mediating the cytoprotective/antioxidative response of rutin in the presence of photooxidative stress. This hypothesis is reinforced by the results obtained in the successive experiments.

As the recent scientific paper [49] demonstrated that the effect of intense UVA radiation (4 kJ/m^2^) for 1–10 min increases cytosolic iron in skin fibroblasts—an event that rapidly leads to cell death by ferroptosis—it seemed important to us to evaluate the involvement of these mechanisms in our experimental system. We observed that a lower radiation supplied (0.06 J/cm^2^), such as the one we used, does not modify the amount of cytoplasmic Fe(II) and Fe(III); on the contrary, it brings out mechanisms of apoptotic type at long-term (2 h).

Therefore, the mechanism we hypothesize about the role of rutin is to activate Nrf2, which not only controls genes for antioxidant enzymes but also prevents apoptosis. Indeed, it has been shown that UVA radiation induces apoptosis through the intrinsic pathway [50], of which Bax is the most important regulatory factor leading to the formation of macropores in the outer mitochondrial membrane and the activation of Caspase-3, the true effector of cell death. Rutin, by increasing Bcl2 and decreasing Bax, leads to an arrest of the apoptotic cascade.

Moreover, Nrf2 is also able to directly modulate mitochondrial functions, including mitophagy and mitochondrial respiration, by forming on the mitochondrial membrane the complex KEAP1/NRF2/PGAM5 (phosphoglycerate mutase 5). In the presence of redox imbalance, the complex on the outer membrane of mitochondria degrades and causes the dissociation of Nrf2. Free Nrf2 enters the nucleus and activates antioxidant and antiapoptotic gene expression to mitigate oxidative mitochondrial stress [51,52,53]. In confirmation of this, experimental evidence shows that cells of Nrf2-KO animals are highly sensitive to chemical-induced mitochondrial damage, whereas chemo-preventive agents protect against mitochondrial damage [54,55]. The results of our study confirm this axiom; in fact, HRR measurements obtained on intact fibroblasts under physiological feeding conditions with a mating control protocol [56,57] revealed that the antioxidant activity exerted by rutin and its ability to activate Nrf2 is also significantly displayed at the mitochondrial level. Indeed, cellular respiration was found to be strongly inhibited in UVA-exposed fibroblasts and maintained at levels comparable to those of the control in rutin-treated cells.

In conclusion, the present study extends our knowledge on rutin’s nutraceutical properties. In fact, although preliminary and worth further analysis and validation, these data demonstrate that rutin is a substance with high photoprotective properties, capable of protecting fibroblasts’ UVA-induced oxidative damage, via the Nrf2 signaling pathway. Nrf2, a transcription factor that regulates the gene expression of a large variety of antioxidants and phase II detoxification cytoprotective enzymes, when upregulated by rutin, maintains both redox homeostasis and the anti-apoptotic response in fibroblasts exposed to UVA rays. Therefore, based on our results, one could hypothesize the use of the rutin as a natural remedy, in food supplements or cosmetic preparations, to protect the skin from the sun, also in the light of issues of environmental sustainability and the circular economy.

## Figures and Tables

**Figure 1 antioxidants-12-00820-f001:**
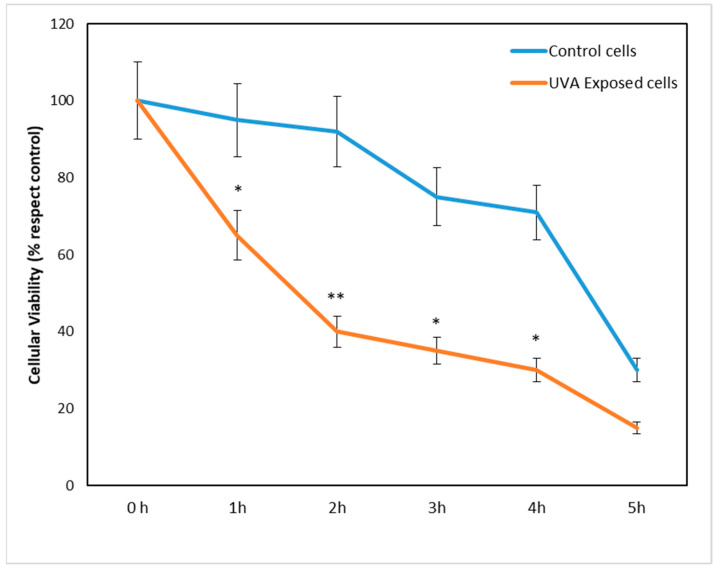
Cell viability after 1, 2, 3, 4, and 5 h of UVA exposure compared with control. Cell viability is expressed as % relative to control (unexposed cells, blue line), which was held for the same time under the same experimental conditions (see details in Section 2). Compared with control fibroblasts, UVA exposure (orange line) produced a significant reduction in cell viability as early as 1 and 2 h. * *p* < 0.05 and ** *p* < 0.01 exposed vs. unexposed.

**Figure 2 antioxidants-12-00820-f002:**
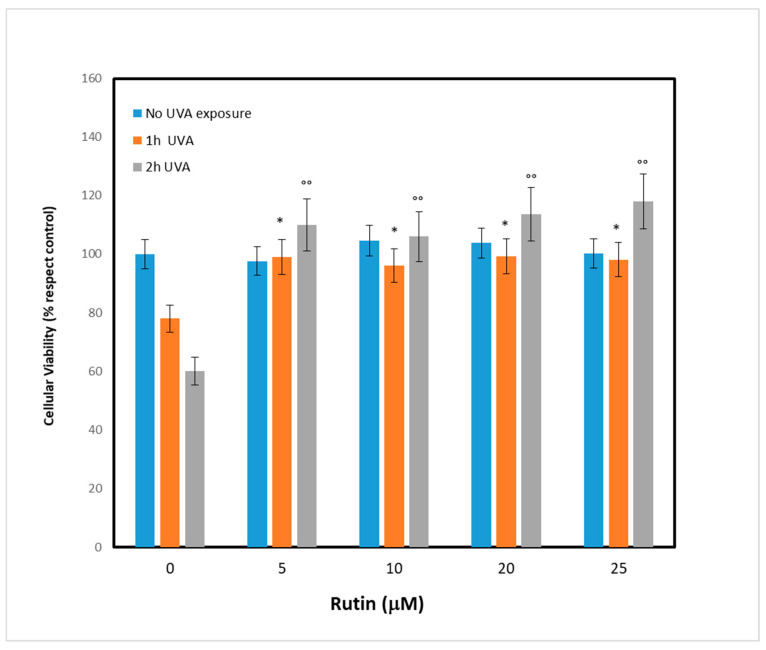
Cell viability in fibroblasts unexposed and exposed to UVA for 1 and 2 h before and after pre-treatment with different concentrations of rutin. Cell viability is expressed as % vs. control (unexposed and untreated cells). Different concentrations of rutin were employed as follows: 0 (absence of pre-treatment), 5, 10, 20, and 25 µM of rutin. Statistical analysis was performed by comparing data under the same experimental conditions (0, 1, and 2 h of UVA exposure). * *p* < 0.05 for 1 h exposure; °° *p* < 0.01 for 2 h exposure.

**Figure 3 antioxidants-12-00820-f003:**
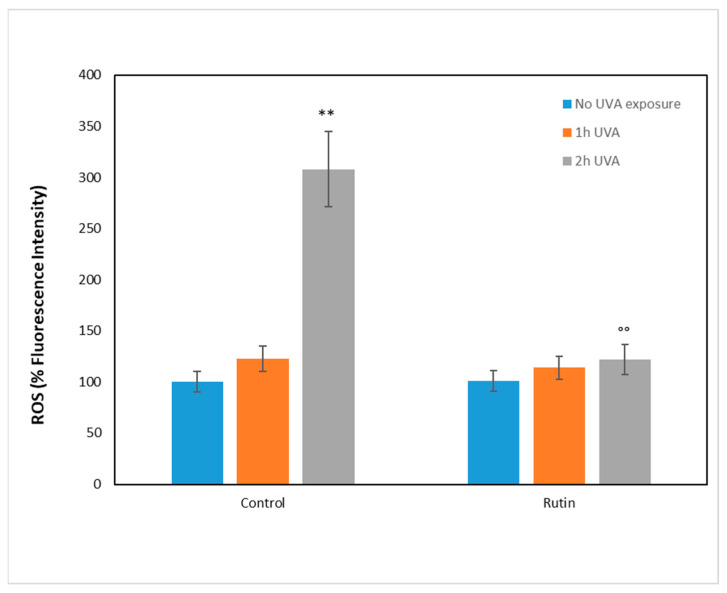
Reactive oxygen species (ROS) for fibroblasts unexposed (blue bars) and exposed to UVA for 1 h (orange bars) and 2 h (gray bars) in the absence (control) and pre-treated with 10 µM of rutin. ROS are expressed as percentage of fluorescence intensity compared with the unexposed and untreated control, arbitrarily considered 100%. Statistical analysis was performed by comparing data between unexposed vs. exposed and between untreated vs. treated fibroblasts. ** *p* < 0.01 unexposed vs. exposed; °° *p* < 0.01 untreated vs. treated.

**Figure 4 antioxidants-12-00820-f004:**
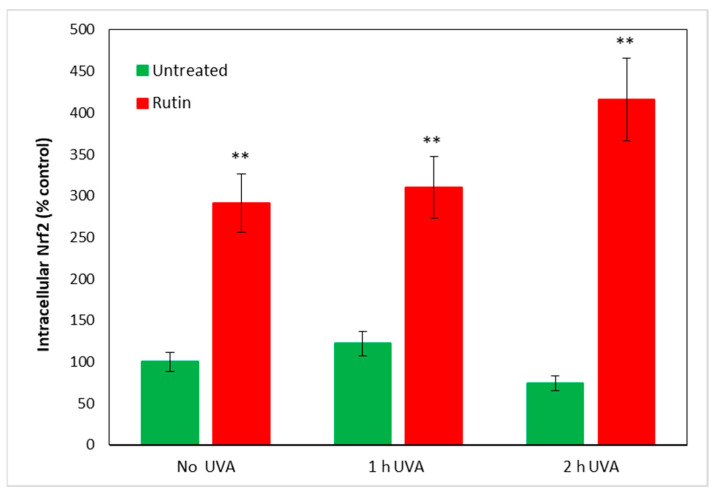
Nrf2 dosage after 1 and 2 h of UVA irradiation with and without 10 µM of rutin. Results are presented as percentage of protein compared with untreated and unexposed control (arbitrarily set as 100%). Rutin induced an increase in Nrf2 levels in unexposed and exposed fibroblasts at both 1 and 2 h of UVA irradiation. ** *p* < 0.01 untreated vs. treated.

**Figure 5 antioxidants-12-00820-f005:**
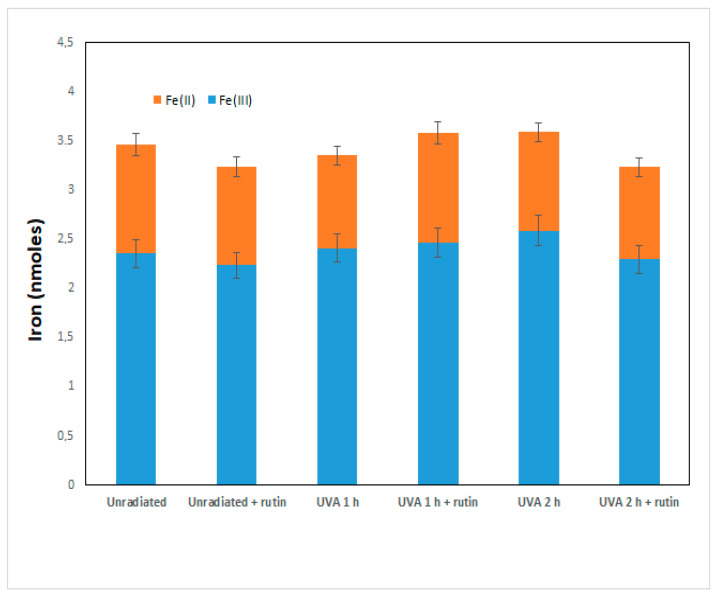
Total iron and oxidized Fe(II) and reduced Fe(III) in the cytosol of fibroblasts unexposed and exposed to UVA and untreated and pre-treated with rutin. Values of iron and its forms are expressed in nmoles and were calculated as reported in the Section 2.

**Figure 6 antioxidants-12-00820-f006:**
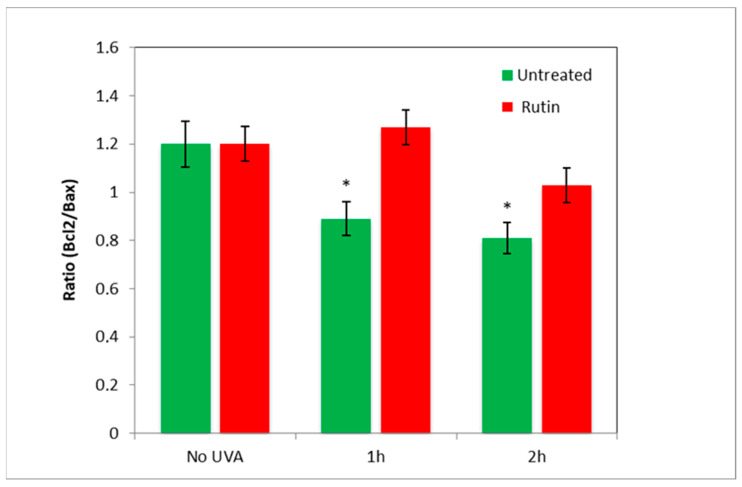
Bcl2/Bax ratio after 1 and 2 h of UVA irradiation with and without 10 µM of rutin. Anti-apoptotic index (expressed as Bcl2/Bax ratio) was calculated in untreated fibroblasts (green bars) and fibroblasts treated with 10 µM of rutin (red bars) unexposed (No-UVA) and exposed for 1 and 2 h of UVA radiation. Pre-treatment with rutin prevented the decrease in the ratio in UVA-exposed fibroblasts. * *p* < 0.05 untreated vs. treated.

**Figure 7 antioxidants-12-00820-f007:**
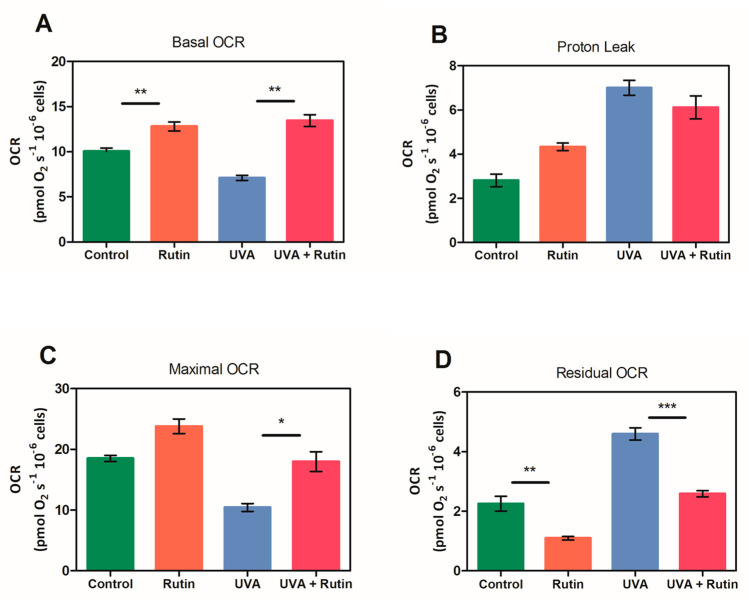
High-resolution respirometry measurements after 2 h of UVA exposure with and without 10 µM of rutin pre-treatment. Measurements were carried out on untreated and unexposed fibroblasts (green bars), fibroblasts treated with 10 µM of rutin (orange bars), those exposed for 2 h to UVA radiation, and those with (red bars) and without (blue bars) pre-treatment with rutin. (**A**) Basal oxygen consumption rate (Basal OCR), (**B**) Proton Leak, (**C**) Maximal OCR, and (**D**) non-mitochondrial respiration (Residual OCR) are expressed as (pmol/(s × 10^6^ cells)) and are average values ± SD of three independent experiments performed in duplicate. The oxygen consumption rate obtained after addition of 0.5 μM rotenone and 2.5 μM antimycin A (Residual OCR) was subtracted from all other OCRs. * *p* ≤ 0.05, ** *p* ≤ 0.01, *** *p* ≤ 0.001.

**Figure 8 antioxidants-12-00820-f008:**
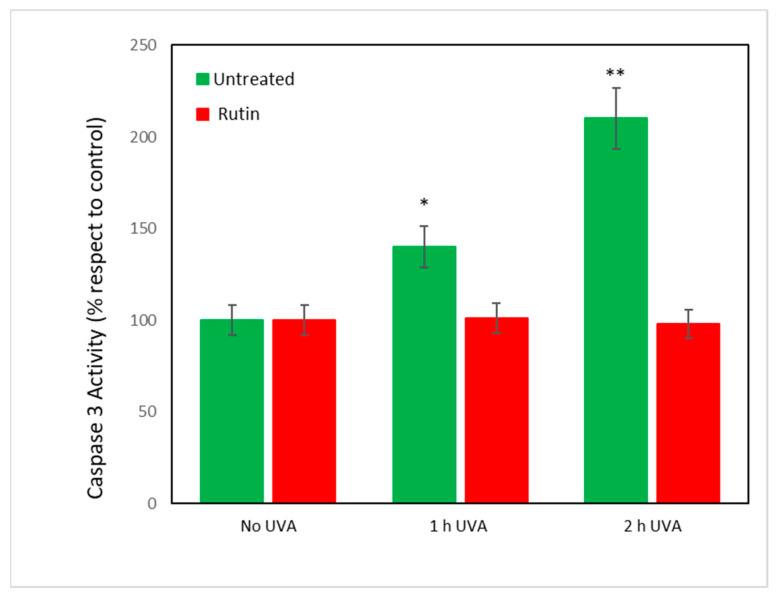
Caspase-3 activity in unexposed and exposed fibroblasts with and without rutin pretreatment. Values are expressed as percentage compared with unirradiated cells (assumed as 100%) in fibroblasts untreated (green bars) and pretreated with rutin 10 μM (red bars) exposed for 1 and 2 h to UVA radiation. Note the protective effect of rutin in irradiated cells with practically unchanged levels of caspase-3 activity. * *p* < 0.05 for 1 h of exposure vs. unexposed and untreated; ** *p* < 0.01 for 2 h of exposure vs. unexposed and untreated.

**Table 1 antioxidants-12-00820-t001:** Total Glutathione (expressed as ng/μL) and Reduced Glutathione (expressed as μg/μL) in fibroblasts in different experimental conditions. The values were calculated as reported in the Section 2. * *p* < 0.05 unirradiated vs. UVA-treated; °° *p* < 0.01 untreated vs. rutin.

	Total Glutathione (ng/µL)		Reduced Glutathione (µg/µL)	
	*Untreated*	*Rutin*	*Untreated*	*Rutin*
**Unirradiated**	57.4 ± 4.5	59.5 ± 7.1	2.30 ± 0.50	5.20 ^°°^ ± 0.62
**UVA 1 h**	57.6 ± 5.5	55.3 ± 4.6	1.36 * ± 0.22	4.63 ^°°^ ± 0.60
**UVA 2 h**	53.3 ± 4.2	59.5 ± 6.5	1.07 * ± 0.20	4.06 ^°°^ ± 0.65

## Data Availability

All data supporting the findings of this study are available within the article.

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
