# Peer review of "Rutin Protects Fibroblasts from UVA Radiation through Stimulation of Nrf2 Pathway"

_antioxidants, 2023, doi:10.3390/antiox12040820_

Round 1

Reviewer 1 Report (Previous Reviewer 1)

The article by Tabolacci et al. has improved considerably with the latest modifications. In addition, they have adequately addressed previous comments, adding further experiments to support their data. The discussion section has also been well expanded. 

I only have one last comment for the authors: is there any data in the literature on the skin permeability of rutin? Does the molecule reach fibroblasts in vivo? Perhaps the authors could say a few words about this in the discussion to further support the beneficial properties of rutin.

Author Response

Reviewer 2 Report (Previous Reviewer 2)

The article has been significantly corrected by the authors. However, some of the results presented in the article seem to be somewhat confusing. For example, in Table 1, the concentration of total and reduced glutathione is presented as ng/µl and µg/µl while the authors declare that it was normalized to protein content. In Figure 5, iron (Fe II and Fe III) is presented as "nmoli" In Figure 2, could the authors compare all results with 0 µM rutin and no UV exposure?”

Author Response

Reviewer 3 Report (Previous Reviewer 3)

the manuscript has been revised accordingly and is now ready for publication.

Round 2

Reviewer 2 Report (Previous Reviewer 2)

The Authors have answered my questions, a minor correction in the legend of Y axis in figure 5 is required (nmoles instead of nmoli).  

This manuscript is a resubmission of an earlier submission. The following is a list of the peer review reports and author responses from that submission.

Round 1

Reviewer 1 Report

The paper by Tabolacci et al. is a well-written paper on the activity of rutin as an Nrf2 inducer in dermal fibroblast, thus preventing the damage due to UVA irradiation.

I have just some minor questions:

- Are the authors sure that the effect on cell viability is really due to a non-reduction of cell number? Since MTS assay follow the NADP(H) enzyme activity and rutin is an inducer of Nrf2 which activation lead to the expression of enzyme with reductase activity, the authors may have observed this kind of secondary effect. Did the authors count cells also? 

- In the sentence lines 219-220 the authors state that an increase in ROS should increase Nrf2. How they explain that after 2 hours of UVA irradiation there is not an increase in Nrf2?

- The authors should expand the discussion paragraph, taking into consideration the recent literature on Nrf2 and the fact that rutin contains in its aglycone an ortho-di-phenol which could be the moiety responsible for the activity (https://doi.org/10.3390/molecules28031356). 

- Rutin should have the capital letter only if at the beginning of a sentence.

Reviewer 2 Report

The work is interesting; however, some data should be better presented;

Generally, fibroblasts used in this experiment should be better characterised. The fibroblasts could be better characterised. In principle in humans, fibroblasts are highly heterogeneous. To characterise a fibroblast cell line, several details should be provided, such as the time from biopsy to the establishment of tissue culture and the body region and age of the donor (Villegas and McPhaul  Curr Protoc Mol Biol 005 Aug;Chapter 28:Unit 28.3. doi: 10.1002/0471142727.mb2803s71. A comparative study made with gene expression of 50 fetal and adult human cultured fibroblasts derived from skin of 16 different body sites displayed distinct transcriptional patterns, suggesting that fibroblasts from various body sites should be considered distinct cell types, which reflect the fact that these cells are probably related to diverse physiological processes (Chang et al. 2002  Proc Natl Acad Sci U S A 2002 Oct 1;99(20):12877-82. doi: 10.1073/pnas.162488599.). Such heterogeneity may create interpretation problems when comparing fibroblasts derived from different body parts from various subjects. Thus, it is essential to consider the purpose of each research, and the Authors should follow these requirements. Btw some other papers may be found citing the same number of approval from the Ethics Commission where not only fibroblasts were used: https://www.ncbi.nlm.nih.gov/pmc/articles/PMC9319849/  https://www.ncbi.nlm.nih.gov/pmc/articles/PMC8701878/

All charts could be "normally" described; looking for the meaning of colours in the text is burdensome. Moreover, all results in figures 1 and 2 could also be presented as % of control while the control could be 100% as it is in Figure 3

The methodology of ELISA studies could be better described. It is hard to understand the principle of the assay. The Authors have written: "The samples were examined with a microplate reader at a wavelength of 450 nm after adding the peroxidase-conjugated secondary antibody. The secondary antibody was against against-Nrf2, a rabbit polyclonal antibody, or against GAPDH, a mouse monoclonal antibody. As I understand, has the appropriate substrate for peroxidase attached to the secondary antibody been added? Moreover, how exactly Nrf2/GAPDH ratio has been calculated? Could the authors provide an example for Nrf2 Bax and Bcl2 separately?

Reviewer 3 Report

This paper titled "Rutin protects fibroblasts from UVA radiation through stimulation of Nrf2 pathway" describes the protective role of rutin in UVA-treated fibroblasts.

The manuscript is focused on a specific role of antioxidant compounds, but is limited in terms of innovation.

In particular, Nrf2 pathway is well defined and studied in terms of pharmacology.
Did you evaluate a possible membrane target for Rutin?

Have you evaluated the formation of quinones or the hydrolysis of rutin in the treated sample?

I suggest including stability data for rutin in the medium and a mass analysis of the supernatant to verify the presence of collateral products.